# The Obscure Limitation of Modular Multilingual Language Models

**Muhammad Farid Adilazuarda⋆[1], Samuel Cahyawijaya⋆[2], Ayu Purwarianti[1]**
[1]Institut Teknologi Bandung    [2] HKUST
`faridlazuarda@gmail.com, scahyawijaya@connect.ust.hk`

## Abstract

We expose the limitation of modular multilingual language models (MLMs) in multilingual inference scenarios with unknown languages. Existing evaluations of modular MLMs exclude the involvement of language identification (LID) modules, which obscures the performance of real-case multilingual scenarios of modular MLMs. In this work, we showcase the effect of adding LID on the multilingual evaluation of modular MLMs and provide discussions for closing the performance gap of caused by the pipelined approach of LID and modular MLMs.

## 1 Introduction

Multilingual language models (MLMs) suffer from the capacity limitation problem known as the **curse of multilinguality**, which penalizes the efficiency of MLMs, both in terms of training and inference, for acquiring new languages. Prior works (Pfeiffer et al., 2020; Ansell et al., 2021; Pfeiffer et al., 2022) alleviate the inference inefficiency bottleneck of the curse of multilinguality by introducing modularity in MLMs through language adapters. This modularity allows MLMs to scale the number of parameters with minimal cost on the training and inference speed. One limitation of modular MLMs is that, as shown in Figure 1, the language of the input needs to be known prior to the inference step for selecting the language adapter. Nevertheless, multilingual evaluations of these modular MLMs make an assumption that an ideal language identification is given and use the language metadata provided on the evaluation data to select the correct language adapter. This produces a gap between modular MLMs in the simulated setting and in the real multilingual scenario. In this work, we address the evaluation gap and further discuss how to mitigate the limitation of modular MLMs.

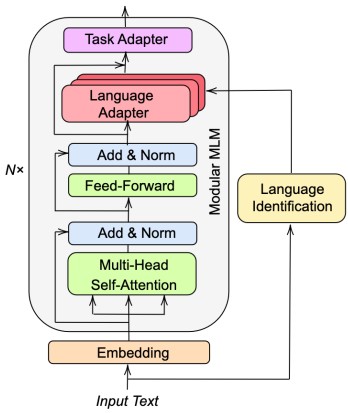

Figure 1: Modular MLMs incorporate language-specific adapters to learn new languages. This renders them language-dependent and reliant on external LID for inference.

## 2 Related Works

**Multilingual Language Model** MLMs (Conneau et al., 2020; Liu et al., 2020; Xue et al., 2021; Workshop et al., 2022) are effective for solving various language understanding and generation in various languages (Hu et al., 2020; Wilie et al., 2020; Cahyawijaya et al., 2021; Adelani et al., 2022; Kumar et al., 2022). To solve the curse of multilinguality of MLMs, the modular MLM approach is introduced. MAD-X (Pfeiffer et al., 2020) and MAD-G Ansell et al. (2021) use adapt MLMs to new languages by using language adapters. X-MOD (Pfeiffer et al., 2022) introduces modularity during pre-training which better aligns modular MLMs across languages.

**Language Identification (LID)** The LID task is introduced over five decades ago (Gold, 1967). Since then, various methods for LID have been introduced, such as n-gram similarity (Cavnar & Trenkle, 1994), naive bayes (Baldwin & Lui, 2010; Lui & Baldwin, 2012; Sites, 2013), and gaussian mixture (Lui et al., 2014). More recently, embedding-based methods using character (Salcianu et al.,

| LID Model | HRL | MRL | LRL | AVG |
|---|---|---|---|---|
| *Fully support languages under study* | | | | |
| FastText | **97.22** | 96.26 | 88.96 | **93.89** |
| CLD3 | 87.84 | 89.30 | **91.47** | 89.57 |
| CLD2 | 76.07 | 90.85 | 85.14 | 83.17 |
| *Partially support languages under study[2]* | | | | |
| langid.py | 92.00 | 93.04 | 76.12 | 86.31 |
| LangDetect | 69.26 | **96.45** | 42.97 | 66.20 |

| NLU Model | HRL | MRL | LRL | AVG |
|---|---|---|---|---|
| *Direct fine-tuning* | | | | |
| XLMR | **86.03** | **84.76** | **83.20** | **84.65** |
| mBERT | 84.76 | 82.50 | 80.62 | 82.64 |
| *Language adapter tuning* | | | | |
| MAD-X (No LID) | 83.30 | 80.96 | 79.46 | 81.27 |
| MAD-X (FastText) | 75.21 | 78.08 | 72.46 | 74.90 |
| MAD-X (CLD3) | 72.90 | 75.20 | 72.89 | 73.47 |

Table 1: Accuracy score of LIDs on MASSIVE. Most LIDs perform well on **HRL** and **MRL**, but the score falls short on **LRL**. **Bold** and underline denote first and second best, respectively.

Table 2: Accuracy score of MLMs on MASSIVE. Incorporating LID decays the performance of the language-adapter model. **Bold** denotes the best performance.

2020) and subwords (Joulin et al., 2017) have also been introduced. In this work, we explore the effect of utilizing these LID modules on the performance of modular MLMs.

## 3 EXPERIMENTAL SETTING

For our experiments, we utilize MASSIVE (FitzGerald et al., 2022), a multilingual intent classification dataset covering 52 typologically-diverse languages. We select 24 languages from MASSIVE and group them into 3 different resource groups based on the language size in CommonCrawl [1] , i.e., high-resource languages (HRL), medium-resource languages (MRL), and low-resource languages (LRL). A detailed list of languages under study and the resource grouping is described in Appendix A. For the LID, we incorporate 5 off-the-shelf LID models, i.e., LangDetect (Nakatani, 2011), langid.py (Lui & Baldwin, 2012), FastText LID (Joulin et al., 2017), CLD2 (Sites, 2013), and CLD3 (Salcianu et al., 2020). We evaluate these LIDs and take the best two LIDs for the multilingual evaluation with unknown languages. For the modular MLM, we utilize MAD-X Pfeiffer et al. (2020) with mBERT backbone. We compare the MAD-X with LID against two direct fine-tuned MLMs and MAD-X without LID. We use accuracy score as the evaluation metric in our experiment.

## 4 RESULT & DISCUSSION

Based on the result of the LID experiment in Table 1, we select FastText and CLD3 for evaluating modular MLMs with unknown languages. The modular MLMs result is shown in Table 2. For the modular MLM without LID, our result aligns with prior works Pfeiffer et al. (2020); Ansell et al. (2021) yielding a slightly lower score compared to the direct fine-tuned models. Both modular MLMs with LID produce an even lower performance in all language resource groups compared to the modular MLM without LID, resulting in a gap of ∼7-8% accuracy score over all language groups. The detailed result of our experiment is shown in Appendix B.

We clearly observe that existing off-the-shelf LID is far from the ideal case which widens the gap to the direct fine-tuning approach and raises an open question for closing the performance gap. To address the question, it is important to understand the limitations of using modular MLMs with off-the-shelf LIDs. Several potential limitations that might occur include: 1) distribution shift of LIDs caused by domain and time differences , 2) label mismatch between LID and the language adapter, and 3) other linguistic problems that affect LIDs such as code-mixing and creole language. We leave the exploration of the solution to these potential limitations for future works.

## 5 CONCLUSION

In this work, we show the limitation of modular multilingual language models (MLMs) in inferencing with unknown languages. We evaluate the effect of using off-the-shelf LID modules on the

---

[1] https://commoncrawl.github.io/cc-crawl-statistics/plots/languages
[2] We zero out the performance for all the unsupported languages.

evaluation of modular MLMs. Our result suggests that using off-the-shelf LID modules significantly decreases the performance of modular MLMs by ∼7-8% accuracy which widens the gap between modular MLMs and non-modular MLMs. In addition, we discuss several potential limitations that might contribute to the performance gap of using off-the-shelf LID with modular MLMs.

URM STATEMENT

All authors of this paper qualify as an underrepresented minority (URM) for the "Tiny Papers" track at ICLR 2023.

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

## A LANGUAGE UNDER STUDY

We provide the list of all languages under study along with the language resource group in Table 3. Language resource is grouped by the size of language data in CommonCrawl, i.e., high-resource languages ($\geq$**1%**, medium-resource languages (($\geq$**0.1%**), and low-resource languages ($<$**1%**).

## B DETAILED EXPERIMENT RESULT

We provide the complete per language result for the language identification and the modular MLMs experiments in Table 4 and Table 5.

---

[3]We use the same number of zh-CN and zh-TW, since there is no Chinese (zh) language variation in CommonCrawl.

| Language | #Speaker | CC Size | Resource Group |
|---|---|---|---|
| ar-SA | 360M | 0.665% | MRL |
| bn-BD | 300M | 0.093% | LRL |
| de-DE | 95M | 5.662% | HRL |
| el-GR | 13.5M | 0.597% | MRL |
| en-US | 373M | 46.320% | HRL |
| es-ES | 493M | 4.435% | HRL |
| fi-FI | 5.4M | 0.398% | LRL |
| fr-FR | 300M | 4.604% | HRL |
| hi-IN | 528M | 0.155% | LRL |
| hu-HU | 13M | 0.599% | MRL |
| hy-AM | 5.4M | 0.032% | LRL |
| id-ID | 300M | 0.781% | MRL |
| is-IS | 0.3M | 0.038% | LRL |
| ja-JP | 128M | 4.532% | HRL |
| jv-ID | 82M | 0.002% | LRL |
| ka-GE | 3.7M | 0.037% | LRL |
| ko-KR | 79.3M | 0.679% | MRL |
| lv-LV | 1.2M | 0.082% | LRL |
| my-MM | 33M | 0.012% | LRL |
| pt-PT | 250M | 1.482% | HRL |
| ru-RU | 258M | 5.717% | HRL |
| vi-VN | 70M | 0.962% | MRL |
| zh-CN | 920M | 4.837% | HRL |
| zh-TW | 4.6M | 4.837%[3] | HRL |

Table 3: List of languages under study in our experiments. The number of speaker information is retrieved from Wikipedia.

| Language | LID-Fasttext | CLD3 | CLD2 | langid | LangDetect |
|---|---|---|---|---|---|
| ar-SA | 94.25 | 86.45 | 81.58 | 91.78 | 94.13 |
| bn-BD | 99.72 | 97.52 | 89.57 | 96.93 | 99.76 |
| de-DE | 97.70 | 88.59 | 89.73 | 92.83 | 82.54 |
| el-GR | 99.68 | 96.91 | 99.77 | 99.84 | 99.64 |
| en-US | 98.61 | 79.44 | 93.43 | 93.96 | 87.82 |
| es-ES | 96.20 | 78.24 | 73.14 | 86.87 | 86.55 |
| fi-FI | 97.70 | 92.91 | 92.90 | 92.08 | 96.09 |
| fr-FR | 98.35 | 87.53 | 85.23 | 94.77 | 94.80 |
| hi-IN | 98.44 | 88.21 | 97.83 | 87.94 | 93.54 |
| hu-HU | 98.54 | 92.24 | 93.89 | 95.34 | 96.71 |
| hy-AM | 99.90 | 98.37 | 99.92 | 99.17 | 0.00 |
| id-ID | 87.20 | 65.86 | 73.54 | 72.68 | 89.32 |
| is-IS | 89.93 | 92.64 | 90.88 | 92.97 | 0.00 |
| ja-JP | 99.41 | 96.63 | 99.04 | 99.11 | 96.23 |
| jv-ID | 24.75 | 68.10 | 0.00 | 22.04 | 0.00 |
| ka-GE | 99.56 | 98.49 | 99.95 | 99.65 | 0.00 |
| ko-KR | 99.50 | 98.47 | 99.03 | 99.96 | 99.36 |
| lv-LV | 90.73 | 90.06 | 95.25 | 94.33 | 97.32 |
| my-MM | 99.93 | 96.90 | 99.97 | 0.00 | 0.00 |
| pt-PT | 92.17 | 83.42 | 77.39 | 77.74 | 84.05 |
| ru-RU | 99.27 | 84.48 | 82.35 | 83.79 | 91.32 |
| vi-VN | 98.41 | 95.85 | 97.26 | 98.62 | 99.53 |
| zh-CN | 97.55 | 98.07 | 84.33 | 99.64 | 0.00 |
| zh-TW | 95.76 | 94.19 | 0.03 | 99.31 | 0.00 |
| **Average** | **93.89** | **89.57** | **83.17** | **86.31** | **66.20** |

Table 4: Per language results of language identification evaluation in MASSIVE.

| Language | XLMR | mBERT | MAD-X | MAD-X w/ FastText | MAD-X w/ CLD3 |
|---|---|---|---|---|---|
| ar-SA | 79.32 | 78.35 | 75.72 | 71.92 | 67.79 |
| bn-BD | 83.25 | 80.23 | 78.61 | 76.36 | 74.95 |
| de-DE | 85.54 | 83.59 | 81.81 | 79.49 | 76.90 |
| el-GR | 85.07 | 81.74 | 80.93 | 79.56 | 78.51 |
| en-US | 88.16 | 86.45 | 85.78 | 83.89 | 83.15 |
| es-ES | 86.18 | 84.97 | 82.58 | 80.97 | 76.43 |
| fi-FI | 85.24 | 82.55 | 82.55 | 79.86 | 77.07 |
| fr-FR | 86.48 | 86.11 | 83.69 | 82.35 | 80.03 |
| hi-IN | 84.63 | 82.38 | 80.73 | 78.14 | 72.73 |
| hu-HU | 85.68 | 82.65 | 81.57 | 80.13 | 76.40 |
| hy-AM | 84.23 | 81.20 | 80.43 | 78.78 | 77.91 |
| id-ID | 86.52 | 84.67 | 82.01 | 76.03 | 69.30 |
| is-IS | 84.16 | 82.21 | 80.40 | 71.49 | 73.57 |
| ja-JP | 85.78 | 84.70 | 83.22 | 82.04 | 81.27 |
| jv-ID | 81.20 | 81.57 | 78.58 | 45.70 | 59.68 |
| ka-GE | 79.19 | 75.25 | 73.23 | 70.85 | 70.17 |
| ko-KR | 85.51 | 84.30 | 82.99 | 81.14 | 80.56 |
| lv-LV | 84.73 | 82.18 | 82.08 | 74.58 | 74.95 |
| my-MM | 82.18 | 78.01 | 78.48 | 76.36 | 74.98 |
| pt-PT | 86.35 | 85.27 | 83.59 | 80.56 | 77.77 |
| ru-RU | 86.65 | 83.96 | 83.52 | 81.74 | 75.45 |
| vi-VN | 86.48 | 83.32 | 82.52 | 79.72 | 78.61 |
| zh-CN | 85.41 | 85.24 | 84.23 | 53.09 | 52.69 |
| zh-TW | 83.73 | 82.55 | 81.27 | 52.79 | 52.45 |
| **Average** | **84.65** | **82.64** | **81.27** | **74.90** | **73.47** |

Table 5: Per language accuracy score of multilingual language models in MASSIVE.

