# OpenReview forum: "The Obscure Limitation of Modular Multilingual Language Models"
_ICLR.cc/2023/TinyPapers — Submitted to Tiny Papers @ ICLR 2023_

### Official Review · Reviewer_pesb · 2023-03-20

**Confidence:** 4

**Summary Of Contributions:**

The performance of modular multilingual language models (MLMs) are evaluated on the assumption that it uses a 100%-accurate language identification (LID) module to select the corresponding language adapter, although a real-scenario LID module might not be perfect. This paper shows how the performance of LID affects the overall performance of modular MLMs and discusses how to mitigate this performance gap.

**Rating:**

High Potential (HP): a submission which meets the reviewing criteria and has potential to make an impact on the field

**Strengths And Weaknesses:**

Strengths
- The paper is well-written and easy to read.
- The paper clearly introduces a problem, showcases a proof-of-concept of the problem via experimental results, and provides empirical comparisons with a few baselines including non-modular MLMs and modular MLM with an oracle LID (also known as “No LID”).
- Experimental settings are described in detail.

Weaknesses
- Although one of the two main objectives (i.e., the problem showcase and the mitigation discussion) in this paper is to discuss how to close the evaluation gap caused by an imperfect LID module, unlike the problem showcase, the paper lacks empirical evidence and further elaboration on the limitation of modular MLMs to support the mitigation discussion.

**Suggested Changes:**

- Not everyone is familiar with “the curse of multilinguality” and how it penalizes the efficiency. I would suggest to add a brief explanation about this in the introduction for clarity.
- I would suggest to add a simple table that summarizes the differences between the underlying architectures of the 5 LID models experimented the in the appendix.

---

### Official Review · Reviewer_GqHq · 2023-04-01

**Confidence:** 3

**Summary Of Contributions:**

The authors investigate the use of modular multilingual language models, specifically MAD-X. They explore the effect of adding language identification on the multilingual evaluation of MAD-X and provide discussion around the results obtained..

**Rating:**

High Potential (HP): a submission which meets the reviewing criteria and has potential to make an impact on the field

**Strengths And Weaknesses:**

*Strengths*
- The paper is well writing, with a clear description of the problem and the methodology followed.
- The problem addressed is correctly framed and it is an interesting contribution.

*Weaknesses*
- At the end of the paper, it is not clear when or why we would need to incorporate an off-the-shelf LID module to a modular MLM. Section 1 aims to give a motivation for carrying on this analysis, but later in Table 2, MAD-X with no LID consistently outperforms the two variants with LID. So, at least in the current experimental setup, the benefit (or the need) of including a LID module is not clear.
- Missing details on how the evaluation was performed (I provide more details on "Suggested Changes")


**Suggested Changes:**

- In the abstract, it would be good to briefly explain why the second sentence is true: the problematic would be more clear and also helps to motivate the work done in the paper.
- It would be good to better define what are the "unknown languages" used for evaluation. It seems like the languages used for evaluating (HRL, MRL, LRL) in Table 2 corresponds to those shown in Appendix A, but these languages are seen during training (or at least most of them). So, what do you mean by "multilingual evaluation with unknown languages"?
- In line with the previous point, the explanation in Appendix A on how you cluster languages into HRL, MRL and LRL should be revisited. The condition given for LRL and/or MRL are not correctly defined, i.e., if the group size is <1% and >=0.1% --> is that MRL or LRL?
- Provide some statistics of the train/test splits, it will help to interpret the results.

Typos:
- In abstract: 'performance gap of caused' -> 'performance gap caused'

---

### Meta-Review · Area_Chair_5cny · 2023-04-06

**Recommendation:** Invite to present
**Confidence:** 3

**Metareview:**

This paper explores the limitations of modular multilingual language models on unknown languages and the effect of adding language identification modules to improve performance. The authors provide guidelines to close the performance gap caused by the pipelined approach of LID and MLMs. The paper is well-presented. The hypothesis is supported by nice set of experiments and comparison with a few baselines. The contribution of the paper is valuable. Some details about the evaluation is missing.

**Summary:**

This paper explores the limitations of modular multilingual language models on unknown languages in a well-supported set of experiments. More details of the evaluation is needed.

**Comments And Feedback To The Authors:**

By addressing reviewers' comments and questions, this paper can become a good contribution to the literature.

**Reason For Not Giving A Higher Recommendation:**

The authors should provide more details about the evaluation to clarify questions of the reviewers.

**Reason For Not Giving A Lower Recommendation:**

The paper is well-presented, the hypothesis is well-supported, and the question is an interesting one for the field.

---

### Decision · Program_Chairs · 2023-04-08

Invite to present